# A Review of Machine Learning Techniques for the Classification and Detection of Breast Cancer from Medical Images

**DOI:** 10.3390/diagnostics13142460

**Published:** 2023-07-24

**Authors:** Reem Jalloul, H. K. Chethan, Ramez Alkhatib

**Affiliations:** 1Maharaja Research Foundation, University of Mysore, Mysuru 570005, India; reem1988jalloul@gmail.com; 2Department of Computer Science and Engineering, Maharaja Research Foundation, Maharaja Institute of Technology, Mysuru 570004, India; hkchethan@gmail.com; 3Biomaterial Bank Nord, Research Center Borstel Leibniz Lung Center, Parkallee 35, 23845 Borstel, Germany

**Keywords:** breast cancer, medical images, machine learning, deep learning

## Abstract

Cancer is an incurable disease based on unregulated cell division. Breast cancer is the most prevalent cancer in women worldwide, and early detection can lower death rates. Medical images can be used to find important information for locating and diagnosing breast cancer. The best information for identifying and diagnosing breast cancer comes from medical pictures. This paper reviews the history of the discipline and examines how deep learning and machine learning are applied to detect breast cancer. The classification of breast cancer, using several medical imaging modalities, is covered in this paper. Numerous medical imaging modalities’ classification systems for tumors, non-tumors, and dense masses are thoroughly explained. The differences between various medical image types are initially examined using a variety of study datasets. Following that, numerous machine learning and deep learning methods exist for diagnosing and classifying breast cancer. Finally, this review addressed the challenges of categorization and detection and the best results of different approaches.

## 1. Introduction

The second-leading cause of mortality is breast cancer, which is the cancer type that is most frequently diagnosed. In 2020, more than 2 million new cases of breast cancer were discovered, making it the most diagnosed disease in the world, according to the World Health Organization (WHO). A total of 626,700 women lose their lives to cancer-related conditions every year. Breast cancer is the most common malignancy in women and the second leading cause of death, and if it is not caught early enough, it can be fatal [1]. If the cancer is found before it expands to a size of 10 mm, the patient has an 85% probability of going into complete remission. Cohort studies show that 30% of breast cancer cases are identified when the tumor is 30 mm. Breast cancer is usually detected during screening when the tumor is at least 20 mm in size [2]. Therefore, encouraging early diagnosis of breast cancer is crucial. A positive clinical breast examination (CBE) and breast self-examination (BSE) may warrant early intervention. Healthcare professionals conduct a CBE as part of routine medical examinations to look for breast lesions. BSE also includes a patient physical examination to look for any physical changes and breast appearance. Women may take charge of their health thanks to the BSE technique. The World Health Organization recommends that at-risk women learn more about BSE [3]. Screening procedures are used to create medical images of the breasts. Professionals with human skills, such as radiologists and doctors, typically interpret these images. Medical imaging’s poor diagnostic accuracy is caused by a lack of technological expertise and expertise in analyzing such images.

Breast cancer has two types, benign (not hazardous) and cancer (malignant) and falls into two groups: normal or aberrant. Benign conditions are thought to be non-cancerous or not life-threatening. But occasionally, it might progress to cancer status. Cancer begins with unnatural cell development and has the potential to spread or infiltrate neighboring tissue quickly. The nuclei of malignant tissue are typically substantially larger than those of normal tissue, which can be fatal in later stages. If malignancy is discovered, tissue is often collected from a surgical biopsy, and less frequently from a tiny needle or larger cardiac biopsy for microscopic examination. Examining current investigative data and extracting pertinent information from earlier data are the key components of an early and precise diagnosis of this disease. Machine learning (ML) algorithms and medical imaging support the procedure. Various deep learning and machine learning techniques have been created to help clinicians understand medical images. Over the past ten years, the development of machine and deep learning models has attracted much attention. Pre-trained networks can be used to install models because they are freely available.

Artificial intelligence (AI) has advanced quickly in recent years. AI aids and assists medical professionals in recognizing and forecasting illness risk more quickly and precisely, enabling early disease identification. Artificial intelligence (AI), a rapidly expanding phenomenon, may soon lead to significant developments in various industries, including medical education.

AI techniques have improved medical image processing, computer-aided diagnosis, image interpretation, fusion, registration, segmentation, image-guided therapy, image retrieval, and image analysis. These techniques enhance the ability of scientists and medical practitioners to comprehend how to recognize the genetic changes that will cause disease [4].

## 2. Artificial Intelligence and Machine Learning Techniques

To anticipate, characterize, or respond to an issue, models must be created using Machine Learning (ML), a subsection of Artificial Intelligence (AI) (Figure 1). Machine learning is a term used to describe a group of techniques that “detect patterns in data, use the uncovered patterns to predict future data or other outcomes of interest”. ML approaches learn directly from data, eliminating the need for numerous expert rules or precisely representing every environmental element. Due to its independence from typical ML extraction techniques and capacity to focus on a complicated hierarchy of image attributes, Deep Learning (DL) is a subclass of Machine Learning (ML) and AI. Computers read data from photos and apply deep learning methods to various previously used computer models to make significant gains. Many medical specialties, particularly radiology and pathology, have used these algorithms to fulfill tasks. Their performance has occasionally been on par with that of human professionals. According to [5], DL can extract information from medical images similar to human analysis and offer details on molecular status, prognosis, and therapy sensitivity. Convolutional neural networks, deep Boltzmann machines, and deep neural networks are popular DL methods.

It is possible to distinguish between supervised, unsupervised, and agent-based machine learning techniques.

▪Supervised learning: In supervised machine learning (ML), every issue may be viewed as the learning of a parametrized function, also referred to as a “model”, that maps inputs (i.e., predictor variables) to outputs (i.e., “target variables”). The purpose of supervised learning is to utilize an algorithm to extract the parameters of those functions from the given data. It is possible to think of supervised learning as using logistic and linear regressions. The majority of ML approaches fall within this category. SVM, DT, Clustering-NN, and K-means are examples of supervised machine learning methods.▪Unsupervised learning: ML issues are often far more difficult to solve if the target variables are unavailable. Unsupervised learning uses the common dimensionality reduction and clustering tasks to find correlations or patterns in the data without providing any direction for the “correct” answer.▪Agent-based learning: Between guided and unguided learning: It is a collection of machine learning techniques where learning occurs by replicating the activities and communications of a single autonomous agent or a group of autonomous agents. To effectively learn, carefully determine values (or preferences), and employ inquiry procedures, one must deal with problems that regularly arise in real life. It is necessary to develop generalizable models because these general unsupervised approaches rely on target obtain variables for which there is little information. Only by experimenting can one detect essential parts of the surroundings. In this context, a specific example of a problem with decision-making over time is reinforcement learning.

This Section provides a concise introduction to frequently employed ML approaches.

### 2.1. Decision Tree

Decision trees (DT) are frequently used in “divide and conquer” data classification strategies. With this method, the data can be shown as a tree, with the leaf nodes serving as the labels for the data sample labels and the inside nodes serving as the various qualities [6]. The appropriate data set is identified by moving up and down the tree from root to leaf. Decision trees (DTs) are a class of supervised learning algorithms, and DTs can be used to classify classification and regression problems. The most widely used DT algorithm is C4.5 [7]. Authors [8] compared C4.5 with other DT algorithms.

### 2.2. Random Forests

The most common use of a bagged DT is called a random forest (RF) [9]. It is an ensemble model made up of numerous separately trained DTs. The class with the most votes is chosen as the final classification for the input data by each component DT in an RF model, which makes classification decisions for each component DT individually. Regression models that use RFs can also produce results that are averaged over the results from individual trees. The RF algorithm’s fundamental idea is that a randomly chosen subset of features is selected at each node of each tree; the samples used to train each component tree are determined using bagging, which resamples (with replacement) the initial set of data points.

### 2.3. K-Nearest Neighbor

A data sample is compared to other data samples using a distance metric in the K-Nearest Neighbor (k-NN) algorithm. The distance between two identical data samples can be reduced using a distance metric, and the distance between two data samples can be increased (The typical Euclidean distance is typically used to calculate the separation between two data samples). An equation will provide the Euclidean distance between x and y. This approach is called the nearest k-method [6].

### 2.4. Support Vector Machine: (SVM)

The Support Vector Machine (SVM) is a popular machine-learning technique for issue regression and categorization. SVM was applied in various applications, including cheminformatics [10] and bioinformatics. The SVM classifier builds a model for the classification using training data. A later step is the classification of an unidentified sample. The main idea behind SVM is using hyperplanes to separate diverse groups. In cases where data can be separated linearly, SVM has attained significant precision. SVM output, however, is unable to separate separable data non-linearly. The data can be split linearly after being mapped to a new, high-dimensional space, utilizing kernel functions to tackle this problem. The correct kernel function selection and its parameters are two of the main issues with SVM [11]. The kernel function is a mathematical technique that enables Support Vector Machines (SVMs) to perform a classification of a set of initially one-dimensional data in a ‘two-dimensional’ manner. Generally, a kernel function facilitates the projection of data from a lower-dimensional space to a higher-dimensional space. Linear kernel function is commonly described as:*K* (*x*, *x_i_*) = *x*·*x^T^*

Polynomial Kernel Function: The polynomial kernel function is directional, i.e., the output depends on the direction of the two vectors in low dimensional space. This is due to the dot product in the kernel. The magnitude of the output is also dependent on the magnitude of the vector x_i_ [12].
K (x, x_1_) = (1+ x·Tx_i_)) d, ‘d’ is degree of kernel function

### 2.5. Artificial Neural Networks (ANN) and Deep Learning (DL)

According to [13] the interconnected neuronal biologic network in the human brain is comparable to an artificial neural network (ANN). The most popular ANN for categorizing ANN issues using the backpropagation training strategy is feedback. In a direct-acting ANN, a single neuron’s basic structure is depicted in Figure 2 [14]. In an ANN, a single neuron obtains input from other neurons, multiplies it by the appropriate weight W_ij_, and then uses an activation function to create a weighted output f. (X_j_). In this kind of network, the neurons are arranged into 180 different types of strata. One input layer, one output layer, and numerous hidden layers are common for direct-action ANNs. The training algorithm provides the technique for weight adjustments in Feedforward ANN during preparation for backpropagation. The output layer compares the actual output generated by the ANN to the desired outcome. The difference between the actual (determined) and anticipated (target) outputs is then used to calculate the error. Finally, during the following iteration, the network receives a new transmission of the mistake of changing the weights. “Deep learning” refers to convolutional neural networks and deep neural networks, which are ANNs with several hidden layers.

#### 2.5.1. Convolutional Neural Network (CNN)

CNNs are the most widely used image algorithms. Convolutional neural networks are commonly used in image classification, image recognition, object detection, etc. [15,16]. The classification of visual images is the deep learning task where convolutional neural networks are most frequently used. In addition, CNN is a high-dimensional image dataset, and each image typically comprises thousands of pixels as traits that are complicated and large in volume. CNN is a feed-forward network that can identify an image’s topological properties. CNNs are multilayer perceptron-driven models. The three distinct layers CNNs use—which are more closely related to neural networks—are convolutional, pooling, and fully connected. Every layer performs a different purpose. The convolutional layer has been used as a feature extractor. The fully connected layer uses the extracted function to determine the current entry’s category. Reducing the number of feature maps and network parameters is the function of a pooling layer.

▪Convolutional Layer: Convolutional layers are utilized to produce feature maps using the weight distribution theory and the local connection concept. Local connectivity and weight distribution objectives are used to reduce the number of parameters while maximizing the advantages of the strongly connected local pixel neighborhood and location-independent local image statistics. The weight distribution model looks like this. Each unit (neuron) in a feature map only has a “local connection” to surrounding patches of the feature map at the previous stage thanks to a weight group called a “filter bank”. Each unit has a filter row they share on a feature map. Other feature maps employ different filter banks as well. The weighted sum of each unit serves as the input to the activation function, a nonlinear transformation function. The weighted total of each succeeding unit is sent to the activation function, a nonlinear transformation function. According to [17], the activation function enables the transmitted data to change nonlinearly for subsequent processing steps.▪Pooling layer: The pooling layer combines (semantically) linked convolutional layer features into one using a subsampling technique. A unit within a pooling layer uses a local patch as input from a previous entity map (convolutional layer) to calculate the maximum or average patch value at the output. Reduced representation size and increased robustness lower the parameters needed in later stages.▪Fully connected layer: According to a multilayer perceptron, a classic neural network, the units in this layer are fully connected to all the units in the layer above.

The organization of its physical space significantly influences CNN’s effectiveness and efficiency. How each layer is constructed, what materials are utilized, and how the layers are arranged impacts how quickly and precisely specific tasks may be completed. Ten CNN architectures will sum it up as follows:LeNet-5: LeNet-5 [13] has two convolutional layers and three fully linked layers.According to [15], AlexNet includes five convolutional and three fully linked layers.VGG-16 [18] uses three fully connected layers and thirteen convolutional layers, taking the ReLU from Alex Net.According to [19], Inception-v1 has a 22-layer architecture with 5 M parameters.According to [20], Inception-v3 is a version of Inception-v1 with parameters of 24 M.ResNet-50: A network with 50 layers [21].Thirty-six convolutional layers, according to Xception [22].Inception-V4: According to [23] Inception-V4 consists of a feature extractor and fully connected layers.One hundred and sixty-four layers are deep in Inception-ResNet.According to ResNeXt-50 [24], it comprises five phases, including an identification and convolution block in each stage. Every identity block contains three levels, and every convolution block has three layers [23].

#### 2.5.2. Recurrent Neural Network (RNN)

The RNN is a specific type of neural network with a specific mathematical description. In the sequence-to-sequence problem, both the input and the output have sequential structures. Recurrent neural networks (RNN) can be utilized to address this problem. The RNN unit has a hidden state that allows it to handle sequential data of different sizes. The hidden state of RNN, which holds some information about a sequence, is its primary and most significant feature. The state, which recalls the previous input to the network, is also known as the memory state. It executes the same procedure on all of the inputs or hidden layers to produce the output, using identical variables for each input. Compared to other neural networks, this minimizes the complexity of the parameter set.

The input and output architecture of RNNs is identical to that of other deep neural network architectures (Figure 2). An RNN keeps track of every piece of data over time. Only the ability to remember past inputs makes it helpful for time series prediction, known as a long short-term memory. RNN is very useful for image recognition, face detection, time series forecasting, language modelling and generating text, and speech recognition machine translation. RNNs are formed as:▪**Bidirectional Neural Network (BiNN):** A BiNN is a type of recurrent neural network in which data are input in both directions and output from both directions is combined to create the input. In cases such as NLP tasks and time-series analysis issues, where the context of the input is more crucial, BiNN is helpful.▪**Long Short-Term Memory (LSTM):** based on the read–write–forget principle, which states that given an input of information, a network should read and write just the information that will be most valuable in forecasting the outcome and ignore the rest. Three new gates are added to the RNN to accomplish this. Only the chosen information is transmitted via the network in this way [25].

#### 2.5.3. Deep Convolutional Neural Network

The most widely used type of deep learning model is the deep convolutional neural network, which is utilized for large-scale image recognition tasks, particularly in the study of medical imaging.

The DCNN consists of three main layers: convolutional layer, pooling layer, and fully connected layer.

(1)Convolutional layer: The main component of the DCNN is the convolutional layer, filter or kernel weights represent the layer parameters. The output feature map is created by multiplying each of the receptive fields, which are the small areas formed by the input feature map. If the stride hyperparameter is smaller than the filter size, the convolution is performed in overlapping windows. The stride is the distance between the applications of filters.(2)Pooling layer: Down sampling the input’s spatial dimension is performed by the pooling layer. The primary goals of this layer type are the gradual reduction of the representation’s spatial dimension and the reduction of the parameters and calculations needed by the network. Although many different pooling functions are available, such as average pooling and L2-norm pooling, max pooling is the most popular since it computes the maximum in the input patch.(3)Fully connected layer: A conventional multi-layer perceptron with a SoftMax activation function in the output layer makes up the completely connected layer. Neuronal cells link to every activation in the preceding layer. To categorize the input image using high-level features extracted from convolutional and pooling layers is the goal of the fully connected layer [26].

## 3. Types of Breast Cancer Imaging

Medical imaging, which frequently employs a range of modalities, including MRI, CT, PET, mammography, radiographic ultrasonography, and duplex ultrasound, is the most efficient way to find breast cancer. Medical images help in disease diagnosis, pathological lesion detection, patient therapeutic care, and the assumption of many disorders. Medical image analysis is one application where ML and AI have been quite successful recently. By using image processing and machine learning techniques for the early identification and diagnosis of malignancies, it is now possible to increase the accuracy of breast cancer diagnosis.

### 3.1. Mammography Images

Mammography is a type of medical imaging that uses a low-dose X-ray system and is mainly used to test for breast cancer. It can be used to find malignant tumors inside the breast. Mammograms help diagnose breast cancer in persons with odd symptoms or breast nodules; even while screening, mammography helps determine the cancer risk in women without overt symptoms. According to the American Cancer Society, every woman over 40 should obtain a mammogram once a year. On a mammography, dense breast tissue may appear white or light grey. Mammograms of younger women may be easier to view since they appear to have larger breasts. They may spot added in situ lesions and reduce invasive tumors compared to MRI and ultrasounds. Mammography remains the gold standard for community breast cancer screening [27]. Mammography is the most reliable and accurate screening technique. Mammography is still used with MRI and ultrasound, especially with high breast tissue density, but it cannot replace mammography. Mammograms can be viewed in various ways to give more details before detection or diagnosis. The two most common mammography views are cardio-caudal (CC) and mediolateral oblique (MLO). The breast is squashed between two paddles in the CC view mammography, which is taken horizontally from an upward projection at a C-arm angle of e 0°. This reveals the glandular tissue, the surrounding fatty tissue, and the outermost edge of the chest muscle. The breast is squashed diagonally between the paddles in the MLO view of mammography, which is captured at a 45-degree angle from the side of the C-arm. As a result, more breast tissue is visible than in other views.

#### Mammography Datasets

Several datasets are available to the public, and they differ in terms of their size, resolution, image format, and the types of images they contain (such as Full-Field Digital Mammography (FFDM), Film Mammography (FM), or Screen-Film Mammography (SFM), and the kinds of abnormalities they contain). DDSM breasts, Mini-MIAS, the Curated Breast Imaging Subset of DDSM (CBIS-DDSM), and BCDR are a few examples of public datasets.
The digital database for screening mammography (DDSM) comprises 2620 mammograms scanned from film which were then separated into 43 volumes (Figure 3). For each example, there are four breast mammograms since the Mediolateral Oblique (MLO) and Cranio-Caudal projections were used to photograph each breast side. The dataset includes pixel-level annotations for the suspicious regions and the ground truth. The breast density score for each patient was calculated using the ACR BI-RADS (American College of Radiology Breast Imaging Reporting and Data System). The file for each case also contains information about the size and resolution of each scanned image. JPEG (Joint Photographic Experts Group) format, available in various formats and resolutions, was used for the images.The Curated Breast Imaging Subset of the DDSM (CBIS-DDSM) is an upgraded version of the DDSM that includes bounding boxes for the region of interest (ROI), updated mass segmentation, and decompressed pictures. The data were picked and reviewed by mammographers with the necessary training, and the images are in the Digital Imaging and Communication in Medicine (DICOM) format. The collection is 163.6 GB in size and contains 6775 studies. There were 10,239 images in total, all mammography scans with associated mask images. CSV files are associated with the dataset that includes the patients’ pathological data. A mass training set, a mass testing set, a training set for calcification, and a testing set for calcification make up the dataset’s four CSV files. The mass testing set only includes images for 378 cancers, whereas the dataset consists of images of 1318 tumors. Images for 1622 calcifications are included in the calcification training set, whereas photos for 326 calcifications are included in the calcification testing set.IN Breast: Breast consists of 410 images and 115 cases. In 90 of the 115 cases, there was malignancy in both breasts. The dataset represents the four types of breast illnesses: breast bulk, breast calcification, breast asymmetries, and breast distortions. Images of (CC) and (MLO) views, stored in DICOM format, are included in the dataset. The dataset also offers the breast density score from the Breast Imaging-Reporting and Data System (BI-RADS).Mini-MIAS: The dataset includes ground truth indicators for potential abnormalities and 322 digital films. The collection contains five types of abnormalities: masses, architectural distortion, asymmetry, and normal. Ultimately, 1024 by 1024 pixels) were used as the final resolution for the images. The images are accessible to everyone on the University of Essex’s Pilot European Image Processing Archive (PEIPA).BCD: The BCDR consists of two mammographic repositories:○The BCDR-FM and the Film Mammography-based Repository.○The BCDR-DM, or Full Field Digital Mammography-based Repository.

The BCDR repositories include normal and atypical breast cancer cases and the clinical information needed to treat them. The 1010 cases in the BCDR-FM are split between 998 females and 12 males. Additionally, it contains 104 detected lesions and 3703 mammographic pictures in the two perspectives, MLO and CC, from 1125 investigations.

### 3.2. Ultrasound Images

Another type of medical imaging technique for finding tumors is breast ultrasonography (Figure 4). The comprehensive medical images of the breast produced by ultrasound are created using sound wave techniques. This method is considered suitable and safe for pregnant women who cannot utilize X-rays or CT scans and nursing mothers. Additionally, pregnant women and nursing mothers who cannot use X-rays or CT scans are considered suitable and safe candidates for ultrasound.

#### Ultrasound Dataset

Grayscale images make up the majority of ultrasound (US) images stored in a DICOM format at Baheya Hospital. The three categories of the US dataset are benign, malignant, and normal. A total of 1100 images were initially acquired. After preprocessing, 780 of the dataset’s initial 1080 images remained. The LOGIQ E9 ultrasound system and the LOGIQ E9 Agile ultrasound system are used in the scanning procedure. High-quality radiology, cardiology, and vascular care imaging routinely use these approaches. They create images with a 1280 × 1024 resolution. ML6-15-D linear probe transducers operate at 1–5 MHz (Table 1).

### 3.3. Magnetic Resonance Imaging (MRI)

Along with ultrasound and mammography, magnetic resonance imaging (MRI) is an early cancer cell detection method. Magnetic fields are used in MRI to produce incredibly accurate three-dimensional (3D) transverse images. A considerable radiation dose is required for a human body MRI to provide precise 3D breast images. Therefore, the diseased area changes rather noticeably when we use an MRI, and no malignancy is found that cannot be seen in any other method. Breast MRI is important for the early detection of breast cancer, in part because it provides quick diffusion-weighted imaging and T2- and T1-weighted CE (contrast-enhanced) imaging, which can be used to characterize lesions further. Breast MRI is essential in identifying complications in women at high risk of breast cancer, among other factors.

High risk of developing breast cancer.Evaluation of the staging period.Neoadjuvant chemotherapy (NAC) follow-up.Evaluation of an auxiliary lymph node region when mammography could not identify the primary location.

A breast MRI takes thirty to forty minutes, following best practices and benchmark procedures.

#### MRI Datasets

Breast–MRI–NACT–Pilot dataset: The database for this dataset is 19.5 GB in size and contains 99,058 MRI images for 64 patients.Mouse–Mammary: This dataset has 23,487 Images, the database is 8.6 GB, and there are 32 patients.

### 3.4. Histopathological Images

The standard for diagnosing cancer has not altered, despite the fast advancements in medical technology. The tissues involved in disease are portrayed in microscopic detail in histopathological image analysis (Figure 5). The pathologists’ experience and factors such as fatigue and a decline in brain function impact the histopathological investigation’s lengthy and highly skilled process.

#### Dataset for Breast Cancer Histopathological Images

The majority of studies on BC histopathology image analysis, according to the literature, are based on small datasets that are often not shared with the scientific community. The Break His dataset is introduced in this review. At four distinct magnifications (40×, 100×, 200×, and 400×), 82 patients provided 7909 microscopic photos of breast tumor tissue that were clinically realistic. These images were collected for BreaKHis. It now has 2480 benign samples and 5429 cancerous ones. All information was made anonymous. Hematoxylin and eosin (HE)-stained breast tissue biopsy slides were used to create the samples. Pathologists from the P&D Lab obtained the samples through surgical (open) biopsy (SOB), prepared them for histological analysis, and labeled them.

### 3.5. Thermography Images

Thermal imaging (Figure 6) often called breast thermography, is created under a microscope. This makes it possible to examine how cells, tissues, and organs’ microscopic anatomy relates to their structural and functional makeup. Finding breast changes that might be symptoms of breast cancer is routinely performed using this painless, non-invasive method [28]. A thermal infrared camera that converts infrared light into electrical impulses and displays it as a thermogram can be used to diagnose breast cancer by identifying body sections that suggest an unusual temperature shift. Thermography is the name of this technique. Thermal imaging employs sensitive and high-resolution thermal cameras and is a promising early diagnosis technique. Thermal imaging, in conjunction with artificial intelligence (AI), is a successful technique for identifying early-stage breast cancer and is projected to have very high levels of prediction.

#### 3.5.1. Thermal Camera

The Plank equation illustrates the connection between the body surface’s wavelength, temperature, and radiation. The discharge of electromagnetic waves occurs when a body’s temperature exceeds absolute zero. A gadget is required to detect this wavelength since infrared light has a range of invisible wavelengths to the human eye. One of the better methods for figuring out the wavelength range is to use a thermal camera. Any object warmer than absolute zero will emit infrared radiation, which can be detected using a thermal camera. The wavelengths of infrared light typically lie between the visible and microwave spectrum. The wavelength range of this infrared light is 0.75 Mm to 1000 mm. Infrared breast thermography may be used to identify breast cancer early to boost the survival probability of patients with the disease. Therefore, a slight asymmetry between the left and right breast temperature patterns may indicate a breast anomaly. The interpretation of the asymmetry in breast thermograms depends critically on several textural characteristics.

#### 3.5.2. Thermography Datasets

The Mastology Research with Infrared Image (DMR) database is used for most IR-thermal image research. There are 287 people in the DMR-IR database, ranging in age from 23 to 120; 186 of them are healthy, and 48 have breast cancer.

### 3.6. Positron Emission Tomography (PET)

Breast PET, also known as Positron Emission Tomography, is a medical imaging method used to visualize and identify metabolic activity in breast tissue. This technique involves introducing a small amount of a radioactive tracer, typically a sugar-based compound, into the patient’s bloodstream. The tracer accumulates in regions with higher metabolic activity, such as cancerous cells, emitting positrons (positively charged particles). When these positrons interact with electrons in the tissue, they produce gamma rays, which are then detected by the PET scanner to generate images.

Machine learning plays a crucial role in breast PET imaging, particularly in improving image quality, enhancing cancer detection accuracy, and aiding in diagnosis. However, compared to other imaging techniques, nuclear medicine modalities such as PET or scintigraphy have been considered less effective in evaluating early-stage breast cancer. Additionally, publicly available breast PET datasets are not as abundant as datasets for other medical imaging modalities.

It is important to emphasize that the successful application of machine learning in breast PET heavily depends on access to high-quality and well-annotated datasets. Having access to such datasets is vital for training accurate and reliable machine learning models for breast PET imaging analysis (Figure 7) [29].

## 4. Review on Machine Learning in Breast Cancer

Using the information acquired from medical imaging, machine learning, and deep learning algorithms may recognize, categorize, and diagnose breast cancer. This section discusses how to spot breast cancer using a mammography, ultrasound, MRI, or a thermogram.

### 4.1. Machine Techniques for Mammogram Images

To determine the most accurate technique to categorize mammography pictures based on breast cancer concerns, the authors of most papers utilized different methods to classify machine learning. The articles for mammography images using several machine learning models (SVM classifier, ANN, K-NN, Fuzzy C-Means, and CNN) are gathered and summarized in Table 2.

The author [30] employed an SVM classifier to categorize the mammography images by the traits deduced from the Hough transformation to determine the mammograms’ traits. On 95 clinical pictures, they used the SVM approach, and the findings demonstrate that, without mentioning accuracy, the suggested method efficiently categorizes the problematic classes of mammograms.

With a 93.1% accuracy rate [31], classified tumors into benign or malignant types using SVM on the DDSM dataset.

The authors [32] conducted a simulation experiment using 44 mammography images from the MIAS database and the same premise as the earlier work. They stated that the accuracy of the mass categorization was as high as 95%. The DDSM database’s simulated mammography images demonstrated the method’s 93% accuracy.

K-means clustering technique is used in the K-SVM-based model for cancer diagnostics described by [33] to extract symbolic objects from tumors. K-SVM estimated an accuracy of 97.38% using the WDBC dataset.

In [34], the authors proposed a method for mass detection that combines deep CNN and SVM. The final fully connected layer data was utilized to build a high-level characteristic representation of the image for categorization on mammography spots in the CNN model.

An improved DenseNet neural network model, also known as the DenseNet neural network model, was suggested in a different study [35] for the accurate and reliable categorization of benign and malignant mammography. The mammography images from 2042 Shanxi Medical University hospital cases were utilized to create the dataset. According to the study’s findings, the DenseNet-II neural network model outperforms other models in categorization.

In [36] used the SVM classifier on the Mini-MIAS and INBreast datasets to categorize breast density and mass as abnormal or normal. With an AUC of 0.9325, the results and performance reviews are 99% correct.

The accuracy of the classification of the extracted regions as mass or non-mass by the authors [37] using DDSM was 98.88%.

The classification approach proposed by [38] uses an SVM classifier to distinguish between abnormalities (mass or microcalcification) and (benign or malignant). INBreast and MIAS were employed for this investigation. The outcomes indicated that the precision value was 99%. The AUC score was 0.9990.

Another researcher [39] also developed an SVM classifier to categorize anomalies utilizing fusion functions on MIAS data. According to the findings, the precision value was 93.17%.

With an accuracy of 91.25%, Ref. [40] suggested a method for categorizing cancer as normal and abnormal using SVM algorithms on the MIAS dataset (109 cases).

Using the MIAS dataset [41], employed an ANN classifier to determine whether something was normal or abnormal before classifying the abnormal into benign or malignant conditions. The results of their work are as follows: RBF accuracy (normal/abnormal): 93.98%; Sn value: 97.22%; RBF accuracy (benign/malignant): 94.29%; Sn value: 100%.

To classify tumors into benign or malignant tissue, authors [42] built Fuzzy C-Means (FCM) on DDSM and MIAS datasets, and they verified that accuracy is 87%, sensitivity value (90.47%), and specificity value (84.84%).

In [43] used an associative classifier with fuzzy–ANN for breast tissue and mass classification. According to the performance assessment, the accuracy for the DDSM dataset is 95.11%.

Authors [44] used the IRMA and MIAS datasets to deploy a K-NN classifier that correctly identified ROI as normal or abnormal with an accuracy of 92.881 ± 0.0099, a Sn value of 92.885 ± 0.0099, and an AUC value of 0.9713.

An updated Dense-Net neural network model, also known as the DensNet neural network model, was suggested by a different study [35] for the accurate and reliable categorization of benign and malignant mammography. The authors initially preprocessed the mammography images. The DenseNet neural network model was then enhanced, and a new DenseNet-II neural network model was produced by replacing the first convolutional layer with the startup structure. Finally, the preprocessed mammography datasets were added to the neural network models AlexNet, VGGNet, GoogLeNet, DenseNet, and DenseNet II. The average model’s accuracy is 94.55%.

The study [45] used an ANN classifier on the MIAS dataset (57 pictures, 37 benign, and 20 malignant tumors), and the accuracy of the results was 96.89%.

K-NN was employed by the authors [46] to classify 100 pictures from MIAS and BancoWeb as benign or malignant. The accuracy of the result was 96%.

Ref. [47] developed CNN with an end-to-end algorithm to obtain high-level attributes. The Henan Provincial People’s Hospital’s Department of Radiology provided the mammography data used in this investigation. The accuracy and AUC ratings for the suggested model are higher.

Convolution neural network (CNN) architectures such as Inception V3, ResNet50, Visual Geometry Group networks (VGG)-19, VGG-16, and Inception-V2 ResNet were applied to the MIAS dataset by the authors of [48]. The classification of mammogram breast images produced the following results: For the 80–20 method and the 10-fold cross-validation method, the overall accuracy, sensitivity, specificity, and precision were 98.96%, 97.83%, 99.13%, 97.35%, 97.66%, and 0.995, respectively. The author used techniques on the DDSM dataset and implemented CNN architecture with transfer learning, with results of 92.84% accuracy.

In addition, the authors [49] proposed K-NN on 252 images from Mini-MIAS and DDSM databases to explore the differences between normal and diseased breast tissues. The accuracy of the result was shown to be 91.2%.

The Fuzzy Gaussian Mixture Model (FGMM) is a method proposed by [50] for classifying 300 photos from DDSM as benign or malignant. The accuracy of the method was 93%.

Authors in [51] used a novel deep learning framework for the detection and classification of breast cancer. They classified images to benign and malignant. They are based on the CNN architecture of GoogleNet and VGGNet for classification. The GoogLeNet, VGGNet, and ResNet architecture individually give an average classification accuracy of 93.5%.

**Table 2 diagnostics-13-02460-t002:** A summary of the research publications for using deep learning and machine learning in mammogram images.

References	Dataset	ML Method	Results
[36]	Mini-MIAS INBreast	SVM Classifier	Accuracy of 99% AUC value is 0.933
[52]	DDSM	SVM Classifier	Sn value is 82.4%
[37]	DDSM	SVM Classifier	Accuracy is 98.9%
[38]	MIAS INBreast	SVM Classifier	Accuracy is 99% ± 0.50 AUC value 0.99 ± 0.005
[39]	MIAS	SVM Classifier	Accuracy 93.17%
[40]	MIAS: 109 cases	SVM Classifier	Accuracy value from 68% to 100%
[41]	MIAS	RBFNN classifier	RBF (normal/abnormal) Accuracy is 93.9% Sn value is 97.2% RBF (benign/malignant) Accuracy is 94.3% Sn value is 100%
[53]	Private-1896 cases	GLCM SFFS (sequential floating forward selection) the bilateral CC and MLO view images	Sn-value is 68.8% Sp value is 95.0% The AUC value is 0.85 ± 0.046
[45]	MIAS: 57 images 37 benign and 20 malignant	CNN classifier	Accuracy is 90.9% AUC value is 96.9%
[46]	MIAS -BancoWeb: 100 images	CNN and hybrid of K-means a	Accuracy 96%
[42]	DDSM	Fuzzy C-Means (FCM)	Accuracy is 87% Sn value is 90 to 47% Sp value is 84 to 84%
[49]	252 images from Mini-MIAS -DDSM	KNN	Abnormality detecting: Accuracy is 91.2% AUC value is 0.98 Malignancy detecting: Accuracy is 81.4% AUC value is 0.84
[50]	300 images from DDSM	Fuzzy Gaussian Mixture Model (FGMM)	Accuracy is 93% Sn value is 90% Sp value is 96%
[44]	IRMA-MIAS	k-NN	Accuracy is 92.8% ± 0.009 Sn value is 92.85% ± 0.01 AUC value is 0.971
[54]	DDSM	CNN and transfer learning	Sensitivity of the mass 89.9%
[55]	DDSM, MIAS	LS SVM, KNN, Random Forest, and Naive Bayes	Accuracy 92%
[56]	(Mini-MIAS) DDSM	CNN	The accuracy of 0.936, 0.890, 0.871 on the DDSM, 0.944, 0.915, 0.892 on the Mini-MIAS for normal, benign, and malignant regions
[48]	MIAS	CNN a pre-trained architecture such as Inception V3, ResNet50, Visual Geometry Group networks (VGG)-19, VGG-16, and Inception-V2 ResNet	Overall Accuracy, Sn, Sp, precision, F-score, and AUC of 98.96%, 97.8%, 99.1%, 97.4%, 97.7%, and 0.995, respectively, for the 80–20 method and 98.87%, 97.3%, 98.2%, 98.84%, 98.04%, and 0.993 for the 10-fold cross-validation method, the TL of the VGG16 model is adequate for diagnosis.
[57]	DDSM	CNN	Accuracy 71.4%

### 4.2. Machine Learning Techniques for Ultrasound Images

By analyzing 138 privately-owned instances, the researchers [58] suggested an SVM machine learning approach for discriminating benign and malignant tumors. The findings show an accuracy of 86.96%.

In 283 privately-owned cases, authors [59] employed numerous models (DT, SVM, RF, K-NN) to distinguish between benign and unsettling lesions accurately.

A data collection contained 8145 breast ultrasound images overall. The authors developed a deep convolutional approach with multi-scale kernels and skip connections [60]. This technique consists of two steps: the first is to identify solid nodules, and the second is to detect whether there are malignant tumors in the image.

Three convolutional layers and two linked layers were employed by the authors of [61] to classify 166 private tumors with 292 benign masses as breast cancer. The outcome for accuracy was 83%.

The authors [62] proposed CNN to detach skin, fibro glandular tissue, 3D mass, and adipose tissue from breast ultrasound images. Quantitative criteria used to evaluate segmentation effects, such as precision, recall, and F1 calculations, all topped 80%, confirming the suggested technique’s capacity to identify functional tissue in a breast ultrasound image.

To identify ultrasonic lesions, the authors [63] investigated using a patch-based LeNet, a U-Net, and a transfer learning method that employed a FCN (Fully Convolutional Network).

In [61] suggested a CNN system concentrating on transfer learning that produced a 93.6% AUC on 292 benign and 166 malignancies to distinguish between benign and malignant breast lesions. They set up the CNN VGG19 model, which had been trained on 882 images of ultrasonography breast masses from the ImageNet dataset. The findings were 83.0% accuracy and 82.4% sensitivity using the mean AUC.

On 46 privately-held images, the authors [64] utilized an SVM classifier to find the tumor location. The accuracy of this method was 95%.

Other studies, such as [65] achieved a 94.81% accuracy rate while utilizing SVM algorithms to distinguish between normal and abnormal instances using 169 privately-held cases.

Additionally, Ref. [66] provided SVM for detecting and diagnosing breast masses using 120 privately-obtained pictures, with findings demonstrating a 95.85% accuracy.

CNN is based on Google Net. The authors [67] trained and tested CNN on 7408 ultrasound images on 829 images. Results from the suggested model attained 90% accuracy and 86% sensitivity.

U-net was used by [68] to identify the bulk of 433 private ultrasound images and the results achived 84% accurately (Table 3).

### 4.3. Machine Learning Techniques (MLT) for Thermography

There are many publications that have utilized thermography to identify and categorize breast cancer. A list of thermography articles employing ML and DL methods is shown in Table 4.

According to varied results between 88.10% and 2.5%, some authors in various papers employed SVM to categorize their images [74]. However, the authors used SVM, K-Nearest Neighbor, and Naive Bayes with 40 owned images with an accuracy of 92.4% for breast cancer detection. In contrast, 90.48%, Sn value 87.6%, and Sp value 89.73% were the results of segmentation and detection using DWT for 306 images by [75].

Additionally, Ref. [76] developed a metaclassifier (ANN, DT, Bayesian, ELM, MLP) to classify 1052 thermogram images, and the accuracy of the findings was 76.01%.

Another method is the proposed ROI, ANN, for segmentation on the Mastology Research dataset with an accuracy of 90.17% [77].

Overall, 1056 thermography images were used by researchers in [78] to apply the CNN model for extracting breast features based on biodata, image processing, and image statistics. The accuracy was 98.95%.

Authors have suggested that Convolutional Neural Networks (CNN) [47] create a system that automatically takes thermographic images of the breast and categorizes them as normal or abnormal. It was used for 63 images (35 normal and 28 aberrant). These data were utilized for training and how to assess how well CNN performed in comparison to three classification methods: Bayes Network (BN), Tree Random Forest (TRF), and Multilayer Perception (MLP). The results were produced using a CNN classifier (100% TPR, SPC, and ACC). On the other hand, the authors used transfer learning with various deep learning pre-trained architectures, including AlexNet, Google Net, ResNet-50, ResNet-101, Inception V3, VGG-16, and VGG-19, to classify breast cancer using a fresh dataset. The dataset was randomly divided into 30% for validation and 70% for training, with images cropped to a fixed size of (224_224) or (227_227) pixels. With a sensitivity of 100%, specificity of 82.35%, and balanced accuracy of 91.18%, their findings showed that the VGG-16 Convolutional Neural Network surpassed the competition.

To diagnose breast cancer, Ref. [79] applied multiple models (CNN, SVM, and Random Forest) on more than 1000 images available on Kaggle. Additionally, to obtain these results, the researcher compared the accuracy of the three models (CNN model accuracy was 99.67%, SVM accuracy was 89.84, and RF accuracy was 90.55%).

Researchers [80] used the CNN (U-Net) model using the DMR-IR dataset to automatically separate and extract the breast area from other thermal image regions. They achieved an accuracy of 99.33%, a sensitivity of 100%, and a specificity of 98.67%.

On the other hand, Ref. [81] used various CNN architectures to detect breast cancer, including Resnet18, Resnet34, Resnet50, Resnet152, Vgg16, and Vgg19. On 5604 thermography images, they applied the proposed models, and 2411 obtained healthy images. They came to these conclusions because Resnet18, Resnet34, and Resnet50, the most stable design among these three, exhibited sound results in cancer classification using thermography.

**Table 4 diagnostics-13-02460-t004:** This table summarizes the research articles for using machine learning algorithms in thermography images.

References	Dataset	ML Method	Performance Evaluation
[82]	50 breast images	SVM	Accuracy is 88.1% Sn value is 85.7% Sp value is 90.5%
[83]	40 images	SVM Naive Bayes K-Nearest Neighbor	Accuracy is 92.5%
[75]	306 images	DWT	Accuracy is 90.5% Sn value is 87.6% Sp value is 89.7%
[84]	63 images	Fuzzy c-means ROI SVM	Accuracy is 100%
[85]	63 thermography Images	Bio-inspired Swarm Techniques	Accuracy: 85.71%, 84.12%, 85.71%, and 96.83% for each swarm
[77]	Mastology Research -Dataset	ROI ANN	Accuracy is 90.2% Sn value is 89.34% Sp value is 91%
[78]	Mastology Research Dataset	CNN	Accuracy is 98.95%
[86]	Mastology Research Dataset	DWAN	Sn value: 0.95
[87]	63 thermographic (35 normal and 28 abnormal)	CNN TRF MLP BN	CNN presents better results than TRF, MLP, and BN and the accuracy between (80–100% for CNN)
[88]	DMR-IR	ANN SVM	SVM sensitivity of 76% and specificity of 84% ANN sensitivity of 92% and specificity of 88%
[79]	Images of approximately 150 patients, either with or without breast cancer, totaling over 1000 (Kaggle available)	CNN SVM Random forest	The accuracy that CNN acquired was 99.67% SVM was 89.84% The accuracy that RF obtained was 90.55%
[80]	DMR_IR	CNN U_NET	Accuracy = 99.33% Sensitivity = 100% Specificity = 98.67%

## 5. Discussion

This section highlights the difficulties in combining ML with breast cancer and identifies many future research goals. 

### 5.1. Datasets

Most researchers (48%) employed the mammography picture dataset, despite literature assessments of past studies in the classification and diagnosis of breast cancer. On various datasets, they applied a range of machine learning approaches. DDSM accomplished 32% of the work and 29% by MIAS. Following that, some used histological images (33%), several studies used open-access datasets (BreakHis), 11% of the researchers used ultrasound, and only 4% used thermography images (Figure 8 and Table 5). 

### 5.2. Results

▪It is clear from earlier studies that researchers put a lot of effort into applying various machine learning models, including Support Vector Machine Learning (SVM), Probability Neural Networks (PNN), and K-Nearest Neighbors (KNN). They ran the models on various medical images then compared the outcomes.▪Researchers have proposed convolutional neural networks (CNNs) for early breast cancer detection. Various CNN architectures, including -Resnet18, Resnet34, -Resnet50, -Resnet152, -vgg16, and vgg19, have been used, along with each architecture’s median and interquartile range. The best outcomes were from the resnet34 and resnet50 convolutional neural network designs, with 100% predicted accuracy in blind validation.▪ML and DL methods for breast cancer still have significant limits and challenges that need to be addressed despite the positive findings of the reviewed literature.▪These approaches offer outstanding outcomes in early breast cancer diagnosis and categorization. And as a result of the review, several important issues were found. 

### 5.3. Challenges

The following discussion includes these issues, as well as internal aspects, potential future research areas, and issues:▪DL needs considerable training data because the data set’s size and quality significantly impact the classifier’s effectiveness. But a lack of data is one of the biggest obstacles to using DL in medical imaging. Generating significant amounts of medical imaging data is challenging because eliminating human error takes a great deal of work from experts and one person. Large medical imaging data sets are difficult to construct because annotating the data takes a great deal of time and effort from a single expert and many experts to eliminate human error. The absence of substantial training datasets has made it challenging to construct deep-learning models for medical imaging, which was the first problem we saw in our studies. Most reviewed studies evaluated and assessed these using various datasets that cancer research organizations or clinics privately collected. The main issue with this method is that it is impossible to compare how well such models function across several investigations.▪The absence of benchmarks provided a hurdle and highlighted a lack of flexibility.▪Another issue with specific papers is using data expansion techniques rather than transferring learning to minimize overfitting.▪Techniques for breast cancer categorization using unsupervised grouping: The supervised learning method was used to classify breast cancer in most of the selected primary papers. These strategies have provided superior results when labeled images are used throughout the training. However, finding breast cancer images with precise, medically labeled criteria might be difficult. There are frequently many unidentified medical images available. Despite being useful knowledge sources, many blank labels cannot be used for supervised learning. Therefore, there is a pressing need for a breast cancer categorization model that may be created using several grouping techniques without supervision.▪Methodology of reinforcement learning for breast cancer classification: The fundamental issue is a lack of sufficient breast cancer image examples to depict all types of breast cancer. Creating a machine learning model that simultaneously learns from its surroundings can be difficult. Therefore, systems for identifying breast cancer from medical photos can perform and be more effective when employing a learning-based reinforcement model.▪Reliability of data collection techniques: The robustness issue of various clinical and technical circumstances must be addressed to integrate new datasets gradually. Different image acquisition scanners, lighting configurations, sizes, and views across many picture modalities, and varying presentation aspects of the coloring and enlargement factors, are a few examples of these modifications.▪Despite its significance in medical picture segmentation, the segmentation’s influence still falls short of what is required for practical use.

### 5.4. Future

In addition to what we have already discussed, there is another crucial point for the future. Thermal images or computed tomography (CT) images of breast cancer may be used in the future in place of the traditional image modalities (mammograms, ultrasound, MRI, and histological) to improve the accuracy of classification models for breast cancer. The identical patient must receive MRI or CT scans. Additionally, pictures of all different breast cancer cases will be collected. Boundary pictures should be labeled to classify multi-class breast cancer because they enable researchers to evaluate the efficacy of the recently created multi-class breast cancer classification model. Furthermore, emerging technologies and methods mentioned in this comprehensive review may be used for future medical idea cases to predict other cancer types such as cervical cancer or lung cancer, which also use any type of medical images.

## 6. Conclusions

This paper covers the most recent research on machine learning and deep learning methods for identifying and categorizing breast cancer. This study aimed to thoroughly understand contemporary breast cancer detection and diagnosis and identify and categorize breast cancer from various medical image types. This review concentrated on the most widely used machine learning approaches, including SVM, DT, Nearest Neighbor, Naive Bayesian Networks, ANN, and Convolutional Neural Networks. According to the papers analyzed, one interesting direction is the development of sophisticated systems that use artificial intelligence that can foretell serious medical issues and assist doctors and patients in preventing breast cancer.

## Figures and Tables

**Figure 1 diagnostics-13-02460-f001:**
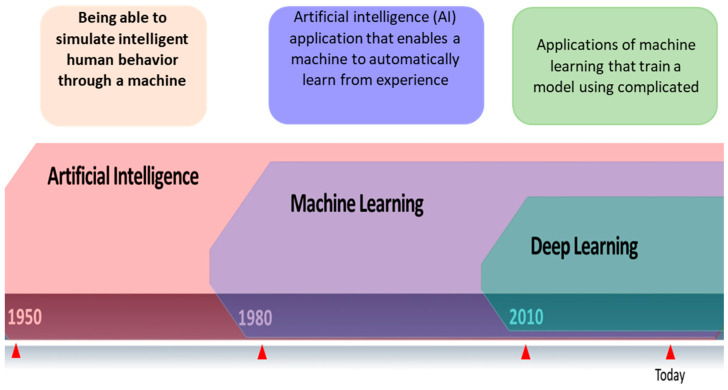
Relationship between artificial intelligence, machine learning, and deep learning.

**Figure 2 diagnostics-13-02460-f002:**
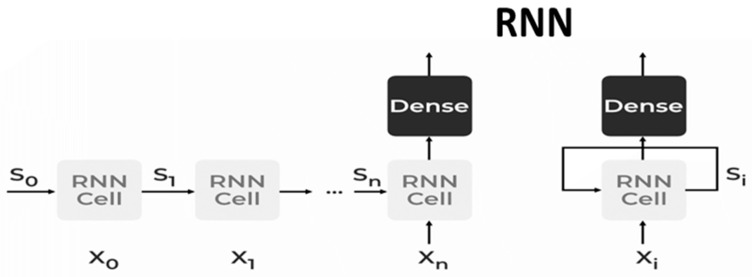
The input and output architecture of RNNs.

**Figure 3 diagnostics-13-02460-f003:**
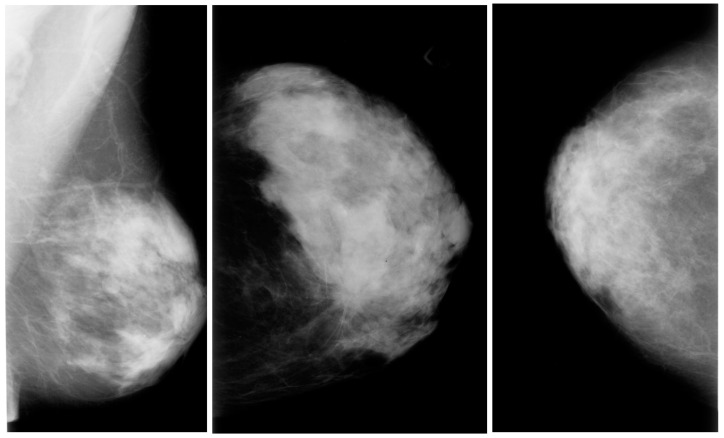
Mammography images from the DDSM dataset from Kaggle.

**Figure 4 diagnostics-13-02460-f004:**
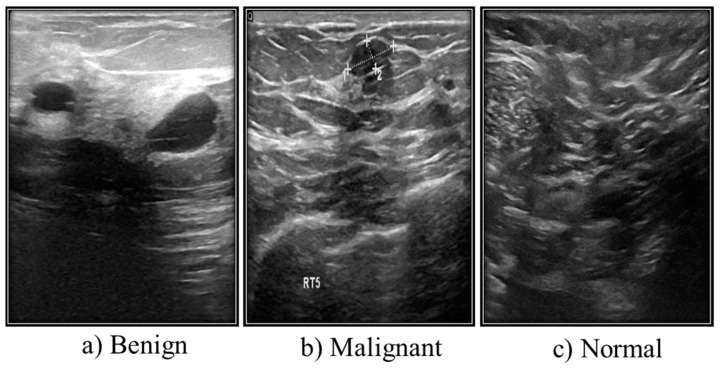
Different types of ultrasound images from Kaggle.

**Figure 5 diagnostics-13-02460-f005:**
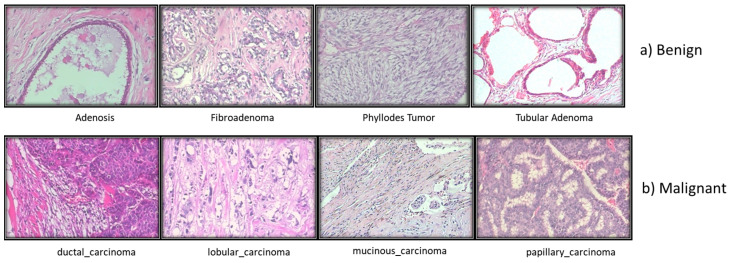
Different types of Breast histopathology images.

**Figure 6 diagnostics-13-02460-f006:**
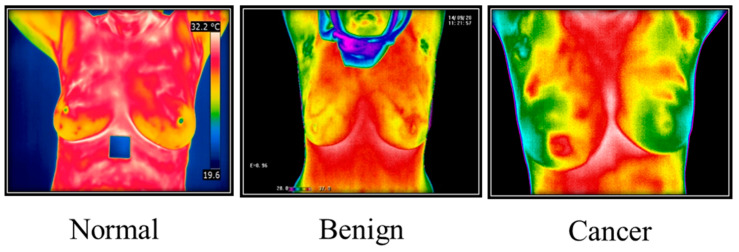
Thermography Images from Irthermo database.

**Figure 7 diagnostics-13-02460-f007:**
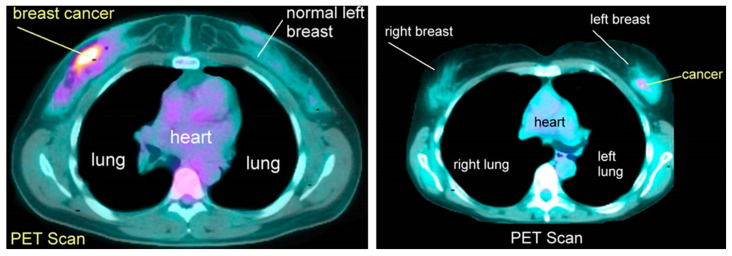
Positron emission tomography scan images.

**Figure 8 diagnostics-13-02460-f008:**
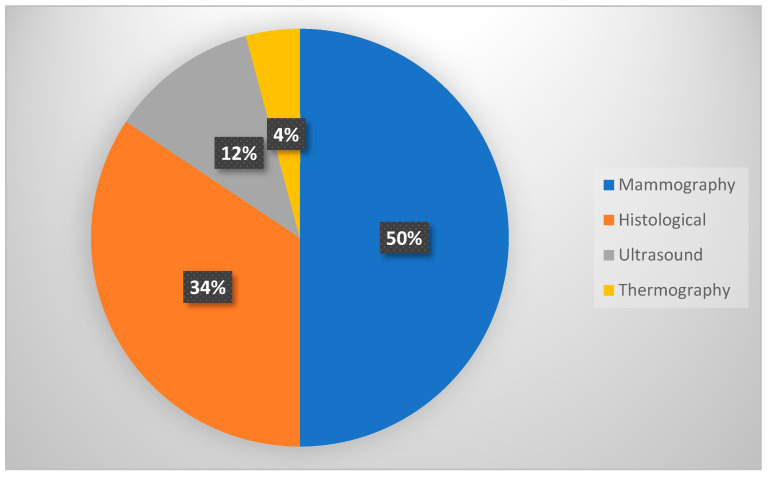
Different types of medical images used.

**Table 1 diagnostics-13-02460-t001:** Table of three classes of ultrasound breast images.

Case	Number of Images
Benign	487
Malignant	210
Normal	133
Total	780

**Table 3 diagnostics-13-02460-t003:** This table summarizes ultrasound papers using different (ML and DL) techniques.

References	Dataset	ML Method	Performance Evaluation
[58]	138 private cases	SVM classifier	Accuracy, Sn value, and Sp value all 86.9% AUC value is 0.89
[66]	120 private images -(benign 70) -(malignant 50)	SVM classifier	Accuracy is 95.85% Sn value is 96.0% Sp value is 91.5% The AUC value is 0.94
[69]	105 private images	SVM classifier	Sn value is 95% Sp value is 90% AUC value is 95%
[65]	169 private cases	SVM classifier	Accuracy is 94.8% Sn value is 94.1% Sp value is 96.7%
[64]	46 private Images	SVM classifier	Accuracy is 0.98 ± 0.013 Sn value is 0.97 ± 0.035 Sp value is 0.98 ± 0.019 AUC value is 0.997 ± 0.003
[70]	97 private images	K-NN	Sn value is 87.8% Sp value is 89.5% AUC value is 0.93
[71]	18 private cases	Binary-LR	Accuracy is 80.4%
[72]	59 private images	RF	AUC value is 99%
[73]	156 owned cases	LR ANN	Accuracy is 81.8% Sn Value is 85.4% Sp Value is 77.8% AUC value is 0.855
[59]	283 owned cases	DT KNN RF SVM	SVM accuracy is 77.7% AUC Value is 0.84 RF accuracy is 78.5% AUC value is 0.83
[67]	7408	CNN based on VGG19	Accuracy value is 91.2% TP value is 84.3% TN value is 96.1% AUC value is 96.0%
[61]	882	CNN based on VGG19	Acc value is 88.7% TP value is 84.8% TN value is 89.7% AUC value is 93.6%
[63]	306	FCN–AlexNet	TP value is 98%
[68]	433	U-Net	TP value is 84%

**Table 5 diagnostics-13-02460-t005:** Different types of available public datasets.

Dataset	Image Type	URL
MIAS	mammogram	https://www.repository.cam.ac.uk/handle/1810/250394 (accessed on 1 June 2023)
DDSM	mammogram	http://marathon.csee.usf.edu/Mammography/Database.html (accessed on 1 June 2023)
mini-MIAS	mammogram	http://peipa.essex.ac.uk/info/mias.html (accessed on 1 June 2023)
Break-His	-histological	https://web.inf.ufpr.br/vri/databases/breastcancer-histopathological-databasebreakhis/ (accessed on 1 June 2023)
DMR-IR	Thermography	http://visual.ic.uff.br/dmi (accessed on 1 June 2023)
BI-RADS	mammogram	https://radiopaedia.org/articles/breast-imaging-reporting-and-data-system-bi-rads (accessed on 1 June 2023)
INbreast	mammogram	http://dx.doi.org/10.17632/x7bvzv6cvr.1 (accessed on 1 June 2023)

## Data Availability

Not applicable.

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
