# Peer review of "A Review of Machine Learning Techniques for the Classification and Detection of Breast Cancer from Medical Images"

_diagnostics, 2023, doi:10.3390/diagnostics13142460_

Round 1

Reviewer 1 Report

The paper of "A Review of Machine Learning Techniques for the Classification and Detection of Breast Cancer from medical images" was investigated in detail. It is interesting study and only there is a few issues which should be solved.

1- Figure 1 is incomplete. Please correct it.

2- ANN in 2.5 section should be explain more. In fact, there are lots of networks which should be mention seperately in the text.

3- There is no "PET" image in the manuscript. Please add in the text.

4- Table 2 is insufficient. The type of networks should be given for ANNs. It is very important information which should be addressed in the paper.

5- The kernel function of SVM methods in previous studies should be added. Please add to the text. 

The paper can be published after these minor corrections.

Minor editing of English language was required.

Author Response

Reviewer 1: We would like to thank the reviewer for careful and thorough reading of this manuscript and for the thoughtful comments and constructive suggestions, which help to improve the quality of this manuscript. 1. Figure 1 is incomplete. Please correct it.: Thank you for your note;(Done) 2. ANN in the 2.5 section should be explained more. There are lots of networks that should be mentioned separately in the text. Response: Thank you for your observation: Multiple techniques of Artificial Neural Networks (ANN) have been added and explained. (RNN, DCNN). 3. There is no "PET" image in the manuscript. Please add in the text. Regrettably, this particular imaging modality has not received adequate attention in research. In comparison to the aforementioned imaging techniques, nuclear medicine imaging modalities, such as Positron Emission Tomography (PET) or scintigraphy, are deemed ineffective in evaluating early-stage breast cancer. Nonetheless, nuclear medicine techniques play a significant role in identifying and categorizing (extra-)axillary lymph nodes, as well as in the staging of distant metastases. Hence, the limited utilization of Deep Learning (DL) in this field of medical imaging is not unexpected, given the aforementioned limitations. 4. Table 2 is insufficient. The type of networks should be given for ANNs. It is very important information that should be addressed in the paper. ANNs have been addressed. 5. The kernel function of SVM methods in previous studies should be added. Please add to the text. The kernel function of SVM methods has been discussed and completed.

Reviewer 2 Report

Overall, this is a well-written manuscript. The author showed a very detailed background and the ML techniques discussed in this article were easy to understand to follow. I don't have any concerns to publish this article.

I think the English is great and easy to understand. I don't have any concerns. 

Author Response

We would like to thank the reviewer for careful and thorough reading of this manuscript and for the thoughtful comments and constructive suggestions.

Reviewer 3 Report

The current manuscript is a survey on breast cancer classification methods. Manuscript should be improved in terms of paper organization and technical details. In this respect, some comments are suggested to be considered in the revised version:

1. Some deep learning-based methods are considered in the manuscript. So, it is better to consider the phrase “machine learning and deep learning” in the title.

2. It is suggested to discuss about the benchmark datasets in this scope with more details (section 5.1).

3. Some papers classified cancer cases in different ranked grades. It is suggested to discuss about it briefly.

4. What is your reference report about Figure 6?

5. Breast cancer classification categorized as medical pattern classification problems. So, it is better to discuss about medical image classification more. For example, I find two papers titled “Developing a Tuned Three-Layer Perceptron Fed with Trained Deep Convolutional Neural Networks for Cervical Cancer Diagnosis”, and titled “Diagnosis of COVID-19 Disease in Chest CT-Scan Images Based on Combination of Low-Level Texture Analysis and MobileNetV2 Features”, which has enough relation. Cite these papers and some other related.

6. Add more technical details about the popular deep networks which are used in this scope. For example, discuss about number of layers, layer types, depth, final layer types, etc.

It is suggested to review whole text in terms of possible English typing errors and grammar mistakes. 

Author Response

Reviewer 3:

We would like to thank the reviewer for careful and thorough reading of this manuscript and for the thoughtful comments and constructive suggestions, which help to improve the quality of this manuscript.

  1. Some deep learning-based methods are considered in the manuscript. So, it is better to consider the phrase “machine learning and deep learning” in the title.

Response: The title has been limited to "Machine Learning Techniques" since Deep Learning is a subset and category within machine learning. Therefore, we have mentioned the general term for the technology instead of the specific term

  1. It is suggested to discuss about the benchmark datasets in this scope with more details (section 5.1)

Response: Thank you for your valuable observation, which highlights one of the main challenges researchers face, namely the absence of a benchmark database. In our research, we will strive to construct and adopt a benchmark database, as it represents a crucial aspect of future work.

  1. Some papers classified cancer cases in different ranked grades. It is suggested to discuss about it briefly.

Response: Thank you for your observation. It's added in mammogram review.

  1. What is your reference report about Figure 6?

Response: Thank you for your observation, the reference added

(Deep and machine learning techniques for medical imaging-based breast cancer: A comprehensive review)

Authors:Essam H. Houssein, Marwa M. Emam, Abdelmgeid A. Ali, Ponnuthurai Nagaratnam Suganthan.

  1. Breast cancer classification categorized as medical pattern classification problems. So, it is better to discuss about medical image classification more. For example, I find two papers titled “Developing a Tuned Three-Layer Perceptron Fed with Trained Deep Convolutional Neural Networks for Cervical Cancer Diagnosis”, and titled “Diagnosis of COVID-19 Disease in Chest CT-Scan Images Based on Combination of Low-Level Texture Analysis and MobileNetV2 Features”, which has enough relation. Cite these papers and some other related.

Response: Thank you for this valuable observation. Uploading and reading research papers is an important step in understanding deep learning techniques and utilizing them in future work We cited these papers in our research paper.

  1. Add more technical details about the popular deep networks which are used in this scope. For example, discuss about number of layers, layer types, depth, final layer types, etc.

Response: We have discussed and added more deep-learning techniques in section (2.5)

Round 2

Reviewer 3 Report

The revised version is better than original submission in terms of technical details and paper organizations. Some of the comments are considered by authors in the revised version. So, minor revision is needed now based on following comments:

1. The title of the sub section 5-4 should be future work ideas. It is suggested to add related references for future medical idea cases such as “cervical cancer and lung cancer detection” which use MRI and CT-scan images too.

2. The runtime of deep learning methods is higher than classical machine learning approaches. So, it is suggested to discuss about the runtime of different methods briefly (based on reported results in related references)

3. I didn’t find any deep neural network architecture in the text (unless RNN). Discuss about deep neural networks with more details about internal layers and differences.  

Author Response

We would like to thank the reviewer for careful and thorough reading of this manuscript and for the thoughtful comments and constructive suggestions, which help to improve the quality of this manuscript.

1. The title of the sub section 5-4 should be future work ideas. It is suggested to add related references for future medical idea cases such as “cervical cancer and lung cancer detection” which use MRI and CT-scan images too.

Thank you for your note;(Done)

2. The runtime of deep learning methods is higher than classical machine learning approaches. So, it is suggested to discuss about the runtime of different methods briefly (based on reported results in related references)

Thank you for your valuable observation. In our future work, we will select one type of medical image and apply different kinds of ML & DL algorithms to it. Then we will certainly compare several features including runtime of different methods.

3. I didn’t find any deep neural network architecture in the text (unless RNN). Discuss about deep neural networks with more details about internal layers and differences.

If I understand your question correctly we have been discussed the three most important main types of neural networks as follows: CNN with ten different types in the section 2.5.1, as well as RNN with their main types (BiNN, LSTM ) in section 2.5.2, and in the section 2.5.3, DCNN has been discussed.
